# Pathogenic *E. coli* in the Food Chain across the Arab Countries: A Descriptive Review

**DOI:** 10.3390/foods12203726

**Published:** 2023-10-11

**Authors:** Mohamed-Yousif Ibrahim Mohamed, Ihab Habib

**Affiliations:** 1Department of Veterinary Medicine, College of Agriculture and Veterinary Medicine, United Arab of Emirates University, Al Ain P.O. Box 1555, United Arab Emirates; 2Department of Environmental Health, High Institute of Public Health, Alexandria University, Alexandria P.O. Box 21511, Egypt; 3ASPIRE Research Institute for Food Security in the Drylands (ARIFSID), United Arab Emirates University, Al Ain P.O. Box 15551, United Arab Emirates

**Keywords:** pathogenic *E. coli*, food safety, foodborne infection, middle east, zoonoses, one health

## Abstract

Foodborne bacterial infections caused by pathogens are a widespread problem in the Middle East, leading to significant economic losses and negative impacts on public health. This review aims to offer insights into the recent literature regarding the occurrence of harmful *E. coli* bacteria in the food supply of Arab countries. Additionally, it aims to summarize existing information on health issues and the state of resistance to antibiotics. The reviewed evidence highlights a lack of a comprehensive understanding of the extent to which harmful *E. coli* genes are present in the food supply of Arab countries. Efforts to identify the source of harmful *E. coli* in the Arab world through molecular characterization are limited. The Gulf Cooperation Council (GCC) countries have conducted few surveys specifically targeting harmful *E. coli* in the food supply. Despite having qualitative data that indicate the presence or absence of harmful *E. coli*, there is a noticeable absence of quantitative data regarding the actual numbers of harmful *E. coli* in chicken meat supplies across all Arab countries. While reports about harmful *E. coli* in animal-derived foods are common, especially in North African Arab countries, the literature emphasized in this review underscores the ongoing challenge that harmful *E. coli* pose to food safety and public health in Arab countries.

## 1. Introduction

Foodborne pathogens continue to pose a significant challenge to both food safety and global trade [1]. Recent times have seen a noticeable increase in the occurrence of harmful *E. coli* strains in the United States of America (USA) [2]. The Centre for Disease Control (CDC) has reported a consistent rise in the number of hospitalizations linked to foodborne illnesses in the USA [3]. In Europe, foodborne infections have emerged as a prominent public health concern [4]. Reports from the European Food Safety Authority (EFSA) and the European Centre for Disease Prevention and Control (ECDC) have designated *Salmonella*, *Campylobacter*, *Listeria*, and Shiga toxin-producing *E. coli* as high-priority pathogens at the European Union (EU) level [5]. The World Health Organization (WHO) [6] has reported that the Middle East and North Africa (MENA) region ranks as the third highest in terms of the burden of foodborne diseases per population, following closely behind the Southeast Asian and African regions. The WHO report also indicates that approximately 70% of foodborne diseases in the Middle East and North Africa region are attributed to *E. coli*, *Campylobacter*, non-typhoidal *Salmonella* (NTS), and Norovirus, underscoring the significant threat posed by these disease-causing agents [6]. Accurately assessing the true prevalence of foodborne infections in the Middle East is a complex task due to limited epidemiological surveillance efforts aimed at identifying individual cases and outbreaks, as well as providing isolates suitable for determining sources and estimating risks at both national and regional levels [7,8]. Moreover, comprehensive data on the extent of antimicrobial resistance in foodborne bacteria responsible for infections in the Middle East region are lacking. While some reports hint at various trends in the relationship between human health and food, a complete understanding of this aspect remains elusive [1,9].

This review offers an updated analysis of the epidemiology of common pathogenic *E. coli* groups that are responsible for causing foodborne illnesses, both in the Arab world and worldwide. The methodology used in this review is a descriptive approach, which involves systematically identifying, evaluating, and compiling a body of literature related to a specific research question. The goal is to identify noticeable patterns or trends related to the research inquiry. Essentially, the descriptive study serves as a crucial part of the analysis, using the collective published information as a comprehensive dataset to draw overarching conclusions regarding the research question [8]. Researchers conducting descriptive review studies extract relevant details from each study, including research methods, publication dates, research findings, and data collection techniques (such as positive, negative, or inconclusive results). These details are then subjected to a recurrence analysis to generate quantitative findings [9]. In conducting this review, various databases, including PubMed, Scopus, Science Direct, Google Scholar, and Web of Science, were employed to gather available research materials on foodborne infections across a wide range of food types in the Arab world over the past two decades. Additionally, efforts were made to identify relevant resources, such as national reports that provide insights into the presence of pathogenic *E. coli* in the food supply chain within the region.

The Arab world encompasses 22 countries situated in the Middle East and North Africa (MENA) region. These countries include Algeria, Saudi Arabia, Sudan, Iraq, Bahrain, Comoros, Djibouti, Egypt, Jordan, Kuwait, the United Arab Emirates, Lebanon, Libya, Mauritania, Morocco, Oman, Palestine, Qatar, Somalia, Syria, Tunisia, and Yemen. Geographically, this region stretches from the Zagros Mountains in Southwest Asia to the Atlantic Ocean, covering a total land area of 14,291,469 square kilometers, which accounts for approximately 10.2% of the world’s total landmass. Within this area, 27.5% is located in Asia, while the remaining 72.5% is in Africa. The region is primarily characterized by dry sub-humid, arid, and semiarid zones, with approximately 90% of the Arab region having limited water resources and arable land, which contributes to its unique ecological profile [10].

In this descriptive review, we aim to elucidate (i) background knowledge and (ii) recent updates, based on published research in the past twenty years, on the prevalence of the concerned pathogenic *E. coli* groups in the food chain in the Arab countries (iii) and the status of antimicrobial resistance (AMR).

*Escherichia coli* is a rod-shaped bacterium with a Gram-negative cell wall structure with dimensions ranging from 1.1 to 1.5 μm in width and 2.0 to 6.0 μm in length; it is a facultative anaerobe that swiftly colonizes the gastrointestinal tracts of both humans and animals shortly after birth, benefiting both the host and bacterium. It belongs to the family Enterobacteriaceae and falls under the *Escherichia* genus. When grown under aerobic conditions at a temperature of 37 °C, it exhibits robust growth on both general and selective agar media. This growth results in the formation of distinct round colonies that produce indole [11]. *E. coli* is typically oxidase-negative, catalase-positive, capable of reducing nitrate to nitrite, shows motility, lacks acid-fast properties, and does not form spores. The identification of specific strains of *E. coli* has traditionally relied on serotyping, a method that involves characterizing the presence of O (somatic), H (flagellar), K (capsular), and F (fimbriae) antigens [12].

In the late 19th century, pediatrician Theodore Escherich discovered *E. coli* and initially referred to it as normal intestinal flora, naming it “*Bacterium coli commune*”. Later, it was officially renamed *E. coli* [13]. This bacterium belongs to the Enterobacteriaceae family and exhibits facultative anaerobic characteristics. *E. coli* can be motile, often utilizing flagella, but can also be non-motile, and it can thrive in both aerobic and anaerobic environments [14]. *Escherichia coli* is one of the most frequently encountered bacteria in clinical samples [15]. In healthy hosts, most *E. coli* strains are non-pathogenic and contribute significantly to the commensal population residing in the host’s intestinal tract, primarily within the mucosal layer of the colon [13]. However, *E. coli* can become pathogenic under certain circumstances, such as when the host’s immune system is suppressed. Some strains are inherently pathogenic and can cause gastrointestinal and urinary tract infections [16]. *Escherichia coli* can endure for extended periods in environmental settings like soil and water [17]. The presence of *E. coli* in food or water can signal inadequate cleaning and careless handling, or it may suggest the potential presence of enteric pathogens [18]. Based on genetic and clinical criteria, *E. coli* can be broadly classified into three major groups: commensal *E. coli*, intestinal pathogenic (diarrheagenic) *E. coli*, and extraintestinal pathogenic *E. coli* (ExPEC). Additionally, various molecular typing techniques, including PCR (polymerase chain reaction) and PFGE (pulsed-field gel electrophoresis), can be employed to differentiate between *E. coli* strains.

The polymerase chain reaction (PCR) has become a cornerstone technique in molecular biology and microbiology for its ability to selectively amplify specific DNA sequences. It is extensively utilized in the detection of genes and their variants within various organisms, including *E. coli*. However, the absence of a PCR signal for a particular gene of interest in *E. coli* isolates can pose challenges and uncertainties. It is crucial to explore the reasons behind such failures and consider the implications they may have on research outcomes and interpretations.

Most *E. coli* found in the environment are non-pathogenic; however, some groups are pathogenic [19,20]. Generally, pathogenic *E. coli* is broadly classified into two major categories: diarrheagenic *E. coli* and extraintestinal pathogenic *E. coli*. The intestinal or diarrheagenic pathogenic strains of *E. coli* are rarely found among the intestinal flora of healthy mammals [18,21]. Based on the virulence factors, six different pathogenic classes of intestinal pathogenic *E. coli* have been identified, namely, enterotoxigenic *E. coli* (ETEC), enteropathogenic *E. coli* (EPEC), enterohemorrhagic *E. coli* (EHEC)/Shiga toxin-producing *E. coli* (STEC), enteroinvasive *E. coli* (EIEC), enteroaggregative *E. coli* (EAEC), and diffusely adherent *E. coli* (DAEC) [13,22]. Extraintestinal pathogenic *E. coli* is phylogenetically and epidemiologically different from diarrheagenic *E. coli*. They could inhabit a range of anatomical locations and cause various infections outside the gastrointestinal tract, among which urinary tract infections are the most common [22]. The *E. coli* strains causing extraintestinal infections have been collectively called extraintestinal pathogenic *E. coli* (ExPEC), which includes two major pathotypes: uropathogenic *E. coli* (UPEC) and neonatal meningitis *E. coli* (NMEC). Several strains of *E. coli* can cause disease in the GIT by toxin production, which includes enterohemorrhagic, enterotoxigenic, enteroinvasive, and enteroaggregative *E. coli* [23,24]. In developing countries, ETEC is the causative agent of travelers’ diarrhea (watery diarrhea without fever). In humans, EIEC is the causative agent of the invasive, dysenteric form of diarrhea because of its ability to invade the colonic mucosa. EHEC produces Vero or Shiga toxins and is the causative agent of hemorrhagic colitis and bloody diarrhea [25,26].

## 2. Epidemiology of Pathogenic *E. coli* Groups

### 2.1. Epidemiology of Enteropathogenic E. coli (EPEC)

Typical enteropathogenic *E. coli* (EPEC) strains are a leading cause of infantile diarrhea in developing countries, whereas they are rare in industrialized countries, where atypical EPEC seems to be a more important cause of diarrhea [27,28]. Also, they are among the most important food-borne pathogens worldwide [29]. EPEC infection results in an excessive loss of water and electrolytes from the body, leading to dehydration and death. However, the underlying molecular mechanisms are not completely understood. EPEC has been reported to disrupt the ion transporters and channels as well as tight junctions in the intestinal epithelial cells leading to the rapid onset of diarrhea. EPEC directly injects virulence factors into the host cells that target multiple signaling pathways of which some have been linked to tight junction disruption [28].

Atypical EPEC strains were obtained from chicken suggesting that it could be a reservoir of these bacteria [29,30]. Their presence in chicken products is evidence that contamination can occur during the slaughtering and manufacturing processes, thereby representing a risk for humans. These results highlight the need for more molecular characterization studies to detect the EPEC genotypes and compare them with human diarrhea. 

EPEC harbors the *eaeA* gene for attaching and effacing or causing A/E lesions on intestinal cells, do not possess the Shiga toxin gene, but may possess other genes such as *bundle-forming pili* (*bfpA*), the intimate adhesin intimin gene. Typical EPEC is *eaeA*-positive and *bfpA*-positive (humans are the reservoirs), while atypical EPEC is only *eaeA*-positive (both humans and animals can be reservoirs).

In Algeria, Ferhat et al. [31,32] successfully isolated the *eaeA* gene from *E. coli* isolates obtained from ovine carcasses in slaughterhouses located in the city of Algiers. Similarly, Chahed et al. [33] and Mohamed et al. [34] also managed to isolate the *eaeA* gene, but this time from bovine carcasses in Algiers slaughterhouses. Furthermore, Dib et al. [35] made a significant discovery, detecting a notable prevalence of the *eaeA* gene in sardines (14.3%, *n* = 32 *E. coli* isolates) and shrimps (33.3%, *n =* 66 *E. coli* isolates). These findings raise awareness about the potential role of sardines and shrimp in the dissemination of the *eaeA* gene within the Algerian context. Interestingly, studies focused on chicken meat in Algeria, such as those conducted by Benameur et al. [36] and Laarem et al. [37], did not find any presence of the *eaeA* gene. These results collectively provide compelling evidence for the contamination of widely consumed bovine and ovine carcasses, as well as fish and seafood (specifically sardines and red shrimp), with EPEC in Algeria. 

In Egypt, several studies have shed light on the role of fresh fish as a potential source of the *eaeA* gene. For example, Galal et al. [38] conducted research in Kafr El-Shikh and identified the *eaeA* gene in 57.1% of 45 *E. coli* isolates obtained from fish samples. Similarly, Saqr et al. [39] found the gene in 83.3% of 6 *E. coli* isolates obtained from Nile tilapia. Fresh beef meat has also been implicated as a source of the *eaeA* gene, with Merwad et al. [40] and Mohammed et al. [41] reporting its presence in 18% (*n =* 27) and 20.7% (*n =* 87) of their respective *E. coli* isolates samples. Additionally, Merwad [40] detected the *eaeA* gene at a rate of 19.1% out of 120 *E. coli* isolates obtained from raw milk samples. The use of poultry waste, sewage, and cow dung as fertilizers for fishponds has been identified as a hazardous source of contamination for water and fish. This contamination poses a direct threat to public health. Detecting EPEC in *E. coli* isolated from fish samples highlights potential risks, as these bacteria are known to cause food poisoning and hemorrhagic enterocolitis in humans who consume improperly processed fish. It is important to note that not all food sources have been found to carry the *eaeA* gene. Hamed et al. [42] were unable to identify the presence of the gene in *E. coli* strains obtained from luncheon and sausage. Similarly, Sahar et al. [43] did not discover the *eaeA* gene in *E. coli* strains collected from a wide range of sources, including minced meat, steaks, sausage, kofta, burgers, luncheon, liver, chicken livers, lambs, oysters, calamari, bivalves, raw milk, yogurt, and cheese. 

In the city of Duhok, Iraq, a study conducted by Taha and Yassin [44] examined various food samples. Out of 120 beef carcass samples, eight isolates tested positive for the presence of the *eaeA* gene. Similarly, from 120 imported chicken carcass samples, two isolates were found to carry the *eaeA* gene. These findings suggest that beef carcasses and imported chicken carcasses could potentially serve as sources for the *eaeA* gene in this region. However, the study did not detect the *eaeA* gene on fish surfaces or in *E. coli* isolated from samples of imported and local raw burgers, local raw ground meat, and local raw milk. These results indicate the absence of the *eaeA* gene in these specific food items. The implications of this study point to beef carcasses and imported chicken carcasses as potential contributors to the dissemination of the *eaeA* gene in the studied area. 

In Jordan, a study conducted by Swedan and Alrub [45] revealed the presence of the *eaeA* gene in 2.8% of 109 isolates obtained from different drinking water sources within Amman city. Additionally, Tarawneh et al. [46] detected the *eaeA* gene in 6% of 50 isolates collected from six slaughterhouses situated in the southern region of Jordan. The detection of the *eaeA* gene within *E. coli* strains from water sources suggests a potential mechanism for the dissemination of virulence genes among various animal species in the environment. This finding underscores the importance of understanding and monitoring the presence of such genes in water sources, as they can contribute to the transmission of virulence factors and impact both animal and public health. Continued research in this area is crucial to comprehending the dynamics of gene spread and potential health risks associated with contaminated water sources in Jordan.

In Lebanon, certain widely consumed dairy products have been identified as potential sources of public health risks due to their role in transmitting the *eaeA* gene. Saleh et al. [47] conducted a study in which they isolated *E. coli* bacteria from a total of 340 dairy products, including Shankleesh, Kishk, and Baladi. The *eaeA* gene was found to be present in 102 *E. coli* isolated obtained from these products. Specifically, the *eaeA* gene was detected in *E. coli* isolated from Kishk and Baladi at rates of 13.5% and 2.7%, respectively. These findings underscore the importance of monitoring and addressing potential contamination of dairy products with pathogenic genes such as *eaeA*. The presence of this gene in certain dairy items highlights the need for robust quality control measures and strict hygiene practices within the dairy production and processing industry. These measures are essential to ensure the safety of these products and protect public health.

In Morocco, the presence of the *eaeA* gene has been identified in various food products, including ground beef, sausage, and turkey. Badri et al. [48] reported that 12.5%, 2.8%, and 2.8% of *E. coli* isolates from 140 ground beef samples, 200 turkey samples, and 120 sausage samples, respectively, tested positive for the *eaeA* gene. The authors emphasized the importance of gaining a better understanding of the molecular characteristics of potentially *eaeA*-positive *E. coli* strains and their role in causing diseases. This understanding is crucial for assessing the pathogenic potential of these strains in human patients. 

In Libya, a study conducted by Garbaj et al. [49] identified the presence of the *eaeA* gene in raw cow milk and raw camel milk, but notably, it was not detected in raw goat milk. These findings highlight the importance of improving and implementing rigorous hygienic practices within the dairy production sector. The authors of the study strongly recommended the adoption and application of Libyan standards for dairy products. This is essential to ensure effective monitoring throughout the entire dairy production process, from the farm to the point of delivery to consumers. 

In Palestine, a study conducted by Adwan et al. [50] reported the absence of the *eaeA* gene in *E. coli* isolated from beef, chicken meat, and turkey samples obtained from the Jenin district. This absence of the gene in these meat products suggests a lower risk of contamination with EPEC carrying the *eaeA* gene in this specific area during the time of the study. It is important to note that the absence of the gene in these isolates contributes to our understanding of the safety of these meat products in the region. However, food safety standards and monitoring should still be maintained to ensure ongoing consumer protection.

In Khartoum, Sudan, a study conducted by Adam [51] did not detect the presence of the *eaeA* gene in *E. coli* isolated from drinking water samples. However, it is important to note that water quality can vary over time and across different sources, so continued monitoring of water sources is essential to ensure the safety of drinking water for the population. While the study did not find the *eaeA* gene in the *E. coli* isolates obtained from samples, maintaining high standards of water treatment and hygiene practices is crucial for public health.

In Qatar, a comprehensive survey conducted by Johar et al. [52] uncovered a high prevalence of the *eaeA* gene in *E. coli* O157:H7 isolated from beef, mutton, and chicken. These findings emphasize the substantial presence of the *eaeA* gene in these food sources, which raises concerns about its potential transmission to humans through contaminated food. This underscores the importance of implementing specialized monitoring programs to detect and control the presence of the *eaeA* gene in food production processes in Qatar.

In Saudi Arabia, a study by Al-Zogibi et al. [53] reported that the primary sources of the *eaeA* gene were milk and raw meat. They isolated *E. coli* from milk, finding it in 15.9% of 540 samples, and detected the *eaeA* gene in 44.2% of the *E. coli* isolates from milk. Similarly, *E. coli* was isolated from raw meat, present in 11.3% of 150 samples, and the *eaeA* gene was detected in 58.8% of the *E. coli* isolates from raw meat. However, it is worth noting that other studies conducted by Hessain et al. [54] and Abu-Duhier [55] did not detect the *eaeA* gene in *E. coli* isolated from beef meats, chicken meats, fresh vegetables, and fruits. These variations in findings could be due to differences in the samples tested, the sampling methods, or regional factors. While the *eaeA* gene was not found in *E. coli* isolated from these specific samples in the studies mentioned, ongoing monitoring and adherence to food safety practices are essential to ensure consumer protection and the safety of food products in Saudi Arabia.

In the United Arab Emirates, the *eaeA* gene was reported in *E. coli* O157 isolated from camel meat, goat meat, cattle meat, and sheep meat obtained from slaughterhouses. This finding indicates the presence of the *eaeA* gene in *E. coli* isolated from animals being processed for meat production. This raises concerns about the potential contamination of meat products intended for human consumption [56] (Table 1).

These studies underscore the importance of continued surveillance and the implementation of hygiene measures in the handling and processing of these meats to mitigate potential risks to public health.

### 2.2. Epidemiology of Enteroinvasive E. coli (EIEC)

Enteroinvasive *E. coli* (EIEC) infections in humans appear to be primarily sourced from infected individuals, as no animal reservoirs have been identified. The main mode of transmission is through the oral–fecal route. While EIEC infections can be found worldwide, they are particularly prevalent in low-income countries where poor general hygiene facilitates their spread. The incidence of Enteroinvasive *E. coli* varies by region [57], and there may be discrepancies in some reports, likely due to the challenge of distinguishing between Shigella and EIEC. In certain countries in Latin America and Asia, such as Chile, Thailand, India, and Brazil, EIEC is frequently identified as a causative agent of diarrhea, with frequent reports of asymptomatic individuals excreting the pathogen. In industrialized countries, EIEC infections are often linked to travel, reported mainly in returning travelers from high-incidence countries. Occasionally, food and water sources have been identified as vehicles of infection, but this is typically traced back to secondary contamination from a human source [58].

Enteroinvasive *E. coli* carrying the *ial* gene can cause sporadic infections but have also been implicated in outbreaks, sometimes affecting a significant number of individuals. In the 1970s, a major outbreak occurred in the United States, impacting 387 patients, and was linked to cheese contaminated with an O124 *E. coli* strain [59]. Europe has also observed an increase in the number of infection cases associated with an emerging EIEC clone. In 2012, Italy reported a large and severe outbreak of bloody diarrhea involving more than 100 individuals [60,61]. During this outbreak, an EIEC O96:H19 strain, a serotype never previously described for EIEC, was isolated, and the suspected source of infection was traced back to cooked vegetables [60]. In the course of the outbreak investigation, an EIEC O96:H19 strain was also found in two asymptomatic food handlers working in the canteen linked to the outbreak. This supported the hypothesis of secondary contamination of the vegetables during post-cooking handling procedures [60]. In 2014, the United Kingdom experienced two interconnected outbreaks of gastrointestinal diseases, affecting more than 100 cases. One of these episodes was associated with the consumption of contaminated salad vegetables, and once again, an O96:H19 EIEC strain was isolated from some of the patients and from vegetable samples [62].

In Arab countries, the presence of the *ial* gene has been a subject of investigation in numerous studies [41,48].

In Egypt, Mohammed et al. [41] detected the *ial* gene in only two *E. coli* O157:H7/H- isolates obtained from beef meat products in Mansoura city. Their study brought attention to the contamination of meat products, especially beef burgers, with various non-O157 STEC and EIEC serotypes. This contamination raised significant concerns regarding potential health risks for consumers of these products.

In Morocco, a comprehensive study conducted by Badri et al. [48] found no detection of the *ial* gene in *E. coli* isolates from ground beef (*n =* 140), sausage (*n =* 120), turkey (*n =* 200), and well water (*n =* 50) (Table 1).

Nevertheless, despite these findings, additional research is warranted to gain a deeper understanding of the prevalence and significance of the *ial* gene markers in *E. coli* strains found in food throughout Arab countries. Such research efforts would contribute to a more comprehensive understanding of the potential implications of the ial gene in public health within the region.

### 2.3. Epidemiology of Enterotoxigenic E. coli (ETEC)

Enterotoxigenic *E. coli* (ETEC) is a major contributor to the global health burden, responsible for an estimated 400 million cases of diarrhea and nearly 400,000 deaths annually among children under 5 years of age in low and middle-income countries [3]. It is also a prevalent cause of travelers’ diarrhea. ETEC is characterized by its ability to produce specific toxins, including heat-labile toxin (LT) and heat-stable toxin (ST), which further comprise two subtypes: STh and STp. ETEC infections are linked to the presence of genes encoding these toxins, namely *elt* (*encoding LT*), *esth* (*encoding STh*), and *estp* (*encoding STp*) [63]. These toxins play a crucial role in causing the diarrheal symptoms and gastrointestinal distress observed in infected individuals. Importantly, ETEC strains can be distinguished by their production of either LT, ST, or both, as well as the specific combination of toxin genes they carry. Additionally, ETEC strains utilize various colonization factors to adhere to the intestinal lining, facilitating the establishment of infection. These factors enhance the bacteria’s ability to colonize and thrive within the host. To date, researchers have identified at least 25 distinct colonization factors associated with human ETEC strains, as detailed in a study by Von Mentzer et al. [64]. Understanding the genetic and molecular factors contributing to ETEC infections is of paramount importance for the development of effective preventive measures. This includes the development of vaccines and the promotion of improved hygiene practices, with the aim of reducing the substantial disease burden caused by ETEC in vulnerable populations.

In Arabic countries, the *elt*, *esth*, and *estp* genes were detected in *E. coli* strains from food samples, specifically in the city of Duhok, Iraq. These *E. coli* strains were isolated from various sources, including beef carcasses (50/20, 41.6%), imported chicken carcasses (52/120, 43.3%), fish surfaces (47/120, 39.1%), imported and local raw burgers (45/120, 37.5%), and local raw ground meat (46/120, 38.3%). The presence of these genes was detected at rates of 34.6%, 91.3%, 100%, 71.4%, and 100%, respectively [44]. This study revealed a high level of food items contaminated with ETEC in the specified area. The authors attributed this high contamination rate to poor hygienic conditions during the slaughtering process, suboptimal food management and storage practices at retail shops, or potential cross-contamination occurring during these processes. Such contamination poses significant community health hazards to the local population and travelers in the region. Therefore, the authors strongly recommend the implementation of strict hygienic practices throughout all stages of food production, handling, and storage to reduce the risk of contamination and subsequent outbreaks of diarrhea. These data provide valuable baseline information for the ongoing monitoring and assessment of ETEC in food products across Arab countries, ultimately contributing to improved food safety and public health in the region (Table 1).

Indeed, various studies have detected the *elt* gene in beef products in different Arabic countries, including Algeria [34] and Egypt [41]. These findings suggest the potential presence of ETEC in beef products in these countries. However, it is worth noting that in some other Arabic countries, there is a lack of published data on the prevalence of ETEC strains carrying the *elt*, *esth*, and *estp* genes in beef products. This underscores the need for further research and surveillance in these regions to better understand the extent of ETEC contamination in various food sources and to implement measures to ensure food safety and public health. Such studies can contribute to a more comprehensive assessment of the situation and help develop strategies for preventing ETEC-related health risks in these areas.

### 2.4. Epidemiology of Enterohemorrhagic E. coli (EHEC) 

Enterohemorrhagic *E. coli* (EHEC) carrying the *stx1*, *stx2*, *eaeA* genes, or Hemolysin (*hlyA*) is a subset of pathogenic *E. coli* capable of causing diarrhea or hemorrhagic colitis in humans [65]. In some cases, hemorrhagic colitis can progress to hemolytic uremic syndrome (HUS) [66], a condition that can lead to acute renal failure in children and result in significant morbidity and mortality in adults [67]. While this has been recognized as a cause of these syndromes since the 1980s, clinical cases and outbreaks attributed to other EHEC serogroups are increasingly being identified [68]. In certain regions, non-O157 EHEC strains may account for a higher number of cases than EHEC O157:H7. What all *E. coli* strains associated with HUS appear to share is the capacity to produce verotoxins and the ability to adhere to and colonize the human intestines. Since verotoxin genes can be transferred between bacteria, it is possible that additional *E. coli* pathotypes linked to HUS could be discovered in the future [69]. According to Yamasaki et al. [70], in Japan, the quantitative detection of shiga toxins directly from stool specimens of patients played a crucial role in identifying an outbreak of EHEC.

The detection of *stx1*, *stx2*, and *eaeA* genes in food samples in various Arabic countries suggests the potential presence of EHEC capable of causing severe health problems. Numerous studies have underscored the presence of EHEC in food products across different regions of the Arab world. Mohammed et al. [41] identified EHEC in ground beef in the city of Mansoura, Egypt, with the presence of EHEC-like strains (*eaeA* + *stx1* or *stx2*). Mohammed et al. [71] detected EHEC in beef products (9.4% of *E. coli* isolates) in Egypt. Merwad et al. [40] found EHEC (5%) in cow milk samples in Egypt. In Iraq, Taha and Yassin [44] discovered EHEC in 30.7% of *E. coli* isolated from 120 beef carcasses. Klaif et al. [72] detected EHEC in camel meat samples from Iraq (45% of *E. coli* isolates). Saleh et al. [47] identified EHEC in dairy products (Kishk and Baladi) in Lebanon at 8.1% and 5.4%, respectively. In Libya, EHEC was detected in *E. coli* O157 from milk and dairy product samples (25%) [49]. In Saudi Arabia, Hessain et al. [54] detected EHEC in raw beef, raw mutton, and raw chicken samples (1%, 2.5%, and 2.5%, respectively). Collectively, these studies reveal a significant contamination of various meat and dairy products with EHEC strains in Arabic countries. The presence of EHEC in these food items underscores a potential health risk for consumers. Consumption of such products can lead to gastrointestinal issues and, in severe cases, conditions like HUS. To mitigate the risk of EHEC contamination and related health hazards, it is essential to ensure strict hygiene practices throughout the entire food production, handling, and distribution processes (Table 1).

### 2.5. Epidemiology of Shiga Toxin-Producing E. coli (STEC) 

Shiga toxin-producing *E. coli* (STEC) carrying *Stx1* and *Stx2* are significant pathogens with global implications, known for their association with various human illnesses. These illnesses include diarrhea, bloody diarrhea, hemorrhagic colitis, and hemolytic uremic syndrome (HUS) [73]. Ruminants, particularly cattle, are recognized as the primary reservoir for STEC, and it can spread to humans through contaminated food and water sources [74]. The severity of STEC infections is influenced by a wide array of virulence factors. One of the key virulence factors is Shiga toxin, which plays a pivotal role in the development of severe symptoms like HUS. Shiga toxin can be categorized into two primary types: Shiga toxin 1 (*stx1*) and Shiga toxin 2 (*stx2*) [73]. These toxins are central to the pathogenicity of STEC and the associated illnesses in humans.

In Algeria, although limited data are available regarding STEC isolated from food sources, several studies have provided valuable insights into the presence and characteristics of STEC strains in different food products. Salih et al. [75] detected a single STEC isolate from frozen bovine meat in Algeria. Dib et al. [35] identified STEC strains in seafood, including three from sardines and three from shrimps. Ferhat et al. [31] conducted a study involving 116 sheep carcasses in an Algiers slaughterhouse. Among the *E. coli* strains isolated from these carcasses, five strains (17.2%) were classified as STEC (Table 1). While the available data are limited, these studies suggest the presence of STEC strains in various food products in Algeria. The detection of STEC in frozen bovine meat, seafood, and sheep carcasses highlights the importance of monitoring and understanding the prevalence of these pathogens in the food supply chain. Continuous research and surveillance efforts are crucial to assess the potential risks associated with STEC contamination in Algerian food sources and to implement appropriate measures to ensure food safety and protect public health.

### 2.6. Epidemiology of Enteroaggregative E. coli (EAEC)

Enteroaggregative *E. coli* (EAEC) infections have been increasingly recognized as important enteropathogens since their initial discovery by patterns of adherence to HEp-2 cells in *E. coli* isolates from Chilean children with diarrhea [76]. EAEC have since been associated with foodborne outbreaks of diarrhea [77], traveler’s diarrhea [78], diarrhea in adults with HIV infection [79], and endemic diarrhea in cities in the USA [80]. A meta-analysis of 41 studies found EAEC to be significantly associated with acute diarrheal illness among both children and adults in developing regions [80]. However, because EAEC are also a highly common infection among children without overt diarrhea in low-resource settings, they have not been found to be a major cause of diarrhea in some endemic settings [81]. Regardless, EAEC, independent of diarrheal symptoms, have been associated with other poor health outcomes in children, such as growth failure [82] and mild-to-moderate intestinal inflammation [76].

**Table 1 foods-12-03726-t001:** The occurrence of virulence *E. coli* genes in foods in Arab countries.

Country	Tested Food Samples (Total Number)	% of *E. coli*-Positive Samples or Isolates No.	Virulence Genes (%) (Out of Total Number) of *E. coli*-Positive Samples	References
**Algeria**	Sheep carcasses (*n =* 363)	ND	*eaeA* (9.92)	[31]
	Bovine carcasses (*n =* 230)	66	*eaeA* (21.2); *stx*1 (10.6); *stx*2 (12.1); *eaeA*, *stx* (4.5)	[33]
	Sardines (*n =* 100)	32	*eaeA* (14.3); *eae*, *stx*1 (14.3); *stx*2 (42.9); *stx*1, *stx*2 (14.3)	[35]
	Shrimps (*n =* 50)	66	*eaeA* (33.3); *stx*1, *stx*2 (16.7); *stx*2 (16.7)	
	Chicken samples (*n =* 32)	56.3	*stx*2 (5.6); *eaeA* (0); *rfbE* (0); *fliC* (0)	[36]
	Retail chicken meat (*n =* 33)	87.8	*stx*1 (6.9); *stx*2 (3.4); *ehxA* (3.4)	[37]
	Frozen beef liver (*n =* ND)	92 isolates	*iss* (85.9); *hylF* (82.6); *ompT* (80.4); *iroN* (87); *fimC* (70.7); *iutA* (90.2); *elt* (5.4); *stx* (2.2); *ipaH* (2.2); *eaeA* (0); *aggR* (0)	[34]
	Chicken samples (*n =* ND)	17 isolates	*iss* (82.4); *hlyF* (52.9) *ompT* (76.5); *iroN* (52.9); *iutA* (52.9); *fimC* (88.2)	[83]
	Frozen bovine meat (*n =* 756)	Five *E. coli* O157:H7 isolates	*stx*1 (20); *stx*2 (100); *eae* (80); *ehxA* (100)	[75]
	Ovine carcasses (*n =* 151)	13 *E. coli* O157:H7 isolates	*eae* (69.2); *stx*1 (7.7); *stx*2 (76.9)	[32]
**Egypt**	Fresh fishes (*n =* 45)	15.6	*eaeA* (57.1); *stx*1 (42.9); *stx*2 (0); *hylaA* (57.1); *sta* (57.1); *Stb* (42.8)	[38]
	Drinking water (*n =* 46)	91	*stx*1 (24.4); *stx*2 (2.4); *eae* (0); *hly* (4.8); *fliCh7* (0)	[84]
	Freshwater canal (*n =* ND)	49 isolates	*eae* (2); *stx*1 (2); *stx*2 (0); *hlyA* (0); *hly* (0)	
	Broiler meats (*n =* ND)	11 isolates	*iroN* (90.9); *ompA* (81.8); *iss* (100); *tsh* (81.8); *papC* (81.8)	[85]
	Karish cheese (*n =* 55)	74.5	*stx* (2.3); *eaeA* (0); *astA* (4.5); *ehaA* (34.8); *lpfA* (33.7); (3.4); *iha* (2.3); *hlyA* (0); *cdt* *cnf* (0)	[86]
	Ras cheese (*n =* 60)	21.7	*stx* (0); *eaeA* (0); *astA* (9.1); *ehaA* (36.4); *lpfA* (45.5); (0); *iha* (0); *hlyA* (4.6); *cdt* *cnf* (4.6)	
	Raw milk (*n =* 120)	19.1	*stx*1(21.7); *stx*2 (34.8); *eaeA* (17.3); *ehxA* (17.3)	[40]
	Fresh beef (*n =* 27)	100	*eae* (18); *ipaH* (18); *stx*1 (18); *stx*2 (10)	[41]
	Beef meat products (*n =* 218)	18.3	*eae* (30); *ipaH* (18); *stx*1(18); *stx*2 (18); *eltB* (8); *estA* (8); *ial* (2)	
	Nile tilapia (Oreochromis niloticus) (*n =* ND)	Six isolates	*eaeA* (83.3); *stx*2 (50); *aadA*2 (50)	[39]
	Minced meat (*n =* 50)	Eight	*eaeA* (12.5); *stx*1 (25); *stx*2 (12.5)	[42]
	Luncheon (*n =* 50)	Four	*eaeA* (0); *stx*1 (0); *stx*2 (0)	
	Beef burgers (*n =* 50)	Two	*eaeA* (100); *stx*1 (0); *stx*2 (0)	
	Sausage (*n =* 50)	10	*eaeA* (0); *stx*1 (20); *stx*2 (0)	
	Karish cheese (*n =* 60)	3.3	*eaeA* (50); *stx*1 (0); *stx*2 (50)	
	Raw bovine milk (*n =* 121)	13.2	*stx*1 (12.5); *stx*2 (18.8); *Sta* (12.5); *lt* (0)	[87]
	Meat products (*n =* 100)	32	*lt* (15.6); *eae* (12.5); *stx*1 (6.3); *stx*2 (9.4); *bfpA* (3.1); *ipaH* (3.1)	[71]
	Drinking water (*n =* 300)	5.3	*lt* (25); *st* (12.5); *stx*1 (18.8); *stx*2 (6.3); *eaeA* (31.3)	[88]
	Raw beef (*n =* 100)	ND	*stx*1 (6); *stx*2 (6)	[89]
	Raw milk (*n =* 100)	ND	*stx*1 (7); *stx*2 (7)	
	Sausages (*n =* 8)	25	*eae* (0); *stx*1 (50); *stx*2 (0); *hlyA* (50); *hly* (0)	[43]
	Kofta (*n =* 6)	33.3	*eae* (0); *stx*1 (50); *stx*2 (50) *hlyA* (50); *hly* (0)	
	Luncheon (*n =* 8)	50	*eae* (0); *stx*1 (0); *stx*2 (0); *hlyA* (0); *hly* (0)	
	Chicken livers (*n =* 6)	50	*eae* (0); *stx*1 (0); *stx*2 (0); *hlyA* (0); *hly* (0)	
	Oysters (*n =* 9)	77.8	*eae* (0); *stx*1 (0); *stx*2 (0); *hlyA* (0); *hly* (0)	
	Calamari (*n =* 7)	57.1	*eae* (0); *stx*1 (0); *stx*2 (0); *hlyA* (0); *hly* (0)	
	Bivalves (*n =* 7)	100	*eae* (0); *stx*1 (0); *stx*2 (0); *hlyA* (0); *hly* (0)	
	Raw milk (*n =* 6)	66.7	*eae* (0); *stx*1 (0); *stx*2 (0); *hlyA* (0); *hly* (0)	
	Yogurt (*n =* 4)	100	*eae* (0); *stx*1 (0); *stx*2 (0); *hlyA* (0); *hly* (0)	
	Cheese (*n =* 4)	75	*eae* (0); *stx*1 (0); *stx*2 (0); *hlyA* (0); *hly* (0)	
	Cheese (*n =* 4)	75	*eae* (0); *stx*1 (0); *stx*2 (0); *hlyA* (0); *hly* (0)	
**Iraq**	Beef carcasses (*n* = 120)	50 (41.6)	*eae* (30.7); *elt* (34.6); *esth* (34.6); *estp* (34.6); *stx*1 (53.8); *stx*2 (53.8); *aggR* (0)	[44]
	Imported chicken carcasses (*n* = 120)	52 (43.3)	*eae* (8.6); *elt* (91.3); *esth* (91.3); *estp* (91.3); *stx*1 (0); *stx*2 (0); *aggR* (8.6)	
	Fish surfaces (*n*= 120)	47 (39.1)	*eae* (0); *elt* (100); *esth* (100); *sstp* (100); *stx*1 (0); *stx*2 (0); *aggR* (0)	
	Imported and local raw burgers (*n* = 120)	45 (37.5)	*eae* (0); *elt* (71.4); *esth* (71.4); *estp* (71.4); *stx*1 (28.5); *stx*2 (28.5); *aggR* (0)	
	Local raw ground meat (*n* = 120)	46 (38.3)	*eae* (0); *slt* (100); *ssth* (100); *sstp* (100); *stx*1 (0); *stx*2 (0); *aggR* (0)	
	Local raw milk (*n =* 120)	43 (35.8)	*eae* (0); *elt* (0); *esth* (0); *estp* (0); *stx*1 (0); *stx*2 (0); *aggR* (0)	
	Fish (*n =* 78)	35.9	*stx1* (89.3)*; stx1* (85.7)*; rfb* (0)	[90]
	Camel meat (*n* = 50)	14	*sta* (100); *uspA* (42); *stb* (0); *stb* (0)	[72]
	Frozen burger (*n* = 50)	7	*sta* (100); *uspA* (42); *stb* (0); *lt* (0)	[91]
	Frozen chicken (*n* = 50)	8	*sta* (62.5); *uspA* (12.5); *stb* (0); *lt* (0)	
	Frozen fish (*n* = 50)	10	*sta* (40); *uspA* (10); *stb* (0); *lt* (0)	
**Jordan**	Drinking water (*n =* ND)	109 isolates	*aat* (12.8); *aaic* (2.8); *eae* (2.8); *ipaH* (1.8); *stx*1(0.9); *stx*2 (0)	[45]
**Lebanon**	Shankleesh (dairy products) (*n =* 340)	28.5	*eaeA* (13.5); *ehly* (8.1); *stx*1 (13.5); *stx*2 (13.5)	[47]
	Baladi (dairy products) (*n =* 340)	66.4	*eaeA* (2.7); *ehly* (5.4); *stx*1 (37.8); *stx*1 (37.8);	
	Kishk (dairy products) (*n =* 340)	7.2	*eaeA* (0); *ehly* (0); *stx*1 (10.8); *stx*1 (10.8);	
	Raw vegetables (*n =* ND)	60 isolates	*stx*1 (0); *stx*2 (0)	[92]
**Morocco**	Ground beef (*n =* 140)	45	*eaeA* (12.5); *aggA* (0); *stx*1 (4.7); *stx*2; (3.1); *lt* (0); *St* (0); *hlyA* (4.7); *Saa* (1.6); *astA* (4.7); *Ial* (0); *ipaH* (0); *iucD* (6.3); *cnf1* (0); *afa* (0); *sfa* (1.6)	[48]
	Sausages (*n =* 120)	30	*eaeA* (2.8); *aggA* (0); *stx*1 (2.8); *stx*2 (0); *Lt* (5.6); *St* (0); *hlyA* (0); *saa* (0); *astA* (27.8); *Ial* (0); *ipaH* (0); *iucD* (16.7); *cnf1* (0); *afa* (0); *pap* (5.6); *sfa* (0)	
	Turkey (*n =* 200)	35.5	*eaeA* (2.8); *aggA* (0); *stx*1 (0); *stx*2 (0); *lt* (0); *st* (1.4); *hlyA* (2.8); *saa* (0); *astA* (19.7); *ial* (0); *ipaH* (8.5); *iucD* (33.8); *cnf1* (0); *afa* (0); *pap* (2.8); *sfa* (0)	
	Well water (*n =* 50)	48	*eaeA* (0); *aggA* (0); *stx*1 (0); *stx*2 (0); *lt* (0); *st* (4.2); *hlyA* (4.4); *saa* (0); *astA* (0); *Ial* (0); *ipaH* (0); *iucD* (4.2); *cnf1* (0); *afa* (4.2); *pap* (0); *sfa* (0)	
	Shellfish (*n =* 82)	6.3	*eaeA* (0);*stx*1 (100); *stx*2 (60)	[93]
	Food products (*n =* 7200)	3.4	*hlyA* (4.3); *pap* (17.1); *sfa* (2.9); *stx*1 (10); *stx*2 (4.3); *eae* (4.3)	[94]
	Ground beef (*n =* 140)	2.1	*stx*1 (100); *stx*2 (66.7); *eaeA* (66.7); *hlyA* (100)	[95]
	Sausage (*n =* 120)	0.8	*stx*1 (100); *stx*2 (0); *eaeA* (0); *hlyA* (0)	
**Libya**	Raw cow’s milk (*n =* 28)	10.7	*vt* (33.3); *eaeA* (33.3)	[49]
	Raw camel’s milk (*n =* 9)	33.3	*vt* (0); *eaeA* (0)	
	Raw goat’s milk (*n =* 7)	28.6	*vt* (100); *eaeA* (100)	
	Fermented cow’s milk (*n =* 28)	25	*vt* (75.7); *eaeA* (75.7)	
	Maasora cheese (*n =* 21)	42.9	*vt* (22.2); *eaeA* (22.2)	
	Ricotta cheese (*n =* 10)	30	*vt* (0); *eaeA* (0)	
**Palestine**	Raw beef (*n =* 300)	44 STEC isolates	*stx*1 (68); *stx*2 (63)	[96]
	Chicken meat (*n* = 15)	100	*vt* (0); *eaeA* (0); *bfpA* (0); *aggR* (6.6); *daaE* (0); *lT* (13.3); *sT* (46.6)	[50]
	Turkey (*n* = 10)	100	*vt* (0); *eaeA* (0); *bfpA* (0); *aggR* (0); *daaE* (0); *lT* (0); *sT* (20)	
**Qatar**	Chickens (*n =* 158)	65 APEC	*ompT* (69)*, hlyF* (69%)*, iroN* (68%); *tsh* (54%); *vat* (4%); *iss* (70%); *cvi/cva* (59%); *iucD* (65%)	[52]
**Saudi Arabia**	Raw beef (*n =* 100)	Two *E. coli* O157:H7 isolates	*stx*1 (100); *stx*2 (100); *eae* (50); *hlyA* (0)	[54]
	Raw mutton (*n =* 40)	One *E. coli* O157:H7 isolate	*stx*1 (100); *stx*2 (100); *eae* (100); *hlyA* (0)	
	Raw chicken (*n =* 40)	One *E. coli* O157:H7 isolate	*stx*1 (100); *stx*2 (100); *eae* (100); *hlyA* (0)	
	Ground beef (*n =* 80)	Four *E. coli* O157:H7 isolates	*stx*1 (75); *stx*2 (75); *eae* (0); *hlyA* (25)	
	Beef burger (*n =* 20)	Two *E. coli* O157:H7 isolates	*stx*1 (50); *stx*2 (0); *eae* (0); *hlyA* (50)	
	Ground chicken (*n =* 20)	One *E. coli* O157:H7 isolate	*stx*1 (100); *stx*2 (100); *eae* (0); *hlyA* (0)	
	Milk (*n =* 540)	15.93	*eaeA* (44.2); *stx*2 (67.4)	[53]
	Raw meat (*n =* 150)	11.3	*eaeA* (58.8); *stx*2 (94.1)	
	Fresh vegetables and fruits (*n =* ND)	16 *E. coli* isolates	*eae* (0); *stx*1 (0); *stx*2 (0)	[55]
**Sudan**	Drinking water (*n =* 184)	46	*IPaH* (12.7); *stx* (6.5); *AggR* (6.5); *eae* (0)	[97]
**United Arab Emirates**	Camel meat (*n =* 140)	4.3 (*E. coli* O157)	*rfbE* (100); *flicH7* (58.3); *hlyA* (75); *uidA* (0); *eaeA* (91.7); *stx2* (100); *stx1* (0)	[56]
	Goat (*n =* 150)	Two (*E. coli* O157)	*rfbE* (100); *flicH7* (0); *hlyA* (50); *uidA* (0); *eaeA* (100); *stx2* (100); *stx1* (0)	
	Cattle (*n =* 137)	1.5 (*E. coli* O157)	*rfbE* (100); *flicH7* (0); *hlyA* (60); *uidA* (0); *eaeA* (60); *stx2* (100); *stx1* (0)	

## 3. Pathogenic Antibiotic Resistance in *E. coli*


*E. coli* is the preferred organism when investigating bacterial resistance levels due to its ability to transfer genetic material not only among its own strains but also to other enteric pathogens [18]. In a study conducted in North Georgia, USA [98], 95 avian pathogenic *E. coli* (APEC) isolates were examined, revealing that 92% of them exhibited resistance to three or more antibiotics. A study by Yuan et al. [99] in China reported that 80% of 71 *E. coli* isolates from the livers of chickens that perished on 10 poultry farms displayed resistance to eight or more antibiotics. Similarly, between 2004 and 2005, Li et al. [100] identified high levels of antibiotic-resistant *E. coli* isolates from diseased chickens in China. These isolates demonstrated complete resistance to tetracycline and trimethoprim/sulfonamide, as well as resistance levels ranging from 79% to 83% to chloramphenicol, ampicillin, ciprofloxacin, and enrofloxacin. The presence of antibiotic resistance in commensal strains of *E. coli* could play a pivotal role in the dynamics of resistance and infectious diseases. European data from France, the UK, and the Netherlands indicate a moderate resistance pattern to ampicillin, streptomycin, tetracycline, and trimethoprim/sulfonamide; low resistance to gentamicin, chloramphenicol, and ciprofloxacin; and no resistance to cephalosporins [101].

Given the widespread detection of pathogenic *E. coli* in various food products (as shown in Table 1), there is a pressing need to enhance national and regional surveillance efforts aimed at monitoring antimicrobial resistance in *E. coli* within our food supply chain. In the primary production of animal-derived foods, there has been a noticeable increase in the use of traditional first-line antibiotics such as sulfonamides, chloramphenicol, ampicillin, tetracycline, and streptomycin. This surge in antibiotic use has led to the emergence and development of antibiotic-resistant *E. coli* due to the selective pressure it exerts [18].

As reported by Hemeg [102] in Saudi Arabia, all recovered pathogenic *E. coli* strains (carrying *stx2* and *eaeA* genes), including those from food samples (20 isolates) and individuals with colibacillosis (100 isolates), exhibited resistance to amoxicillin–clavulanic acid, penicillin, and erythromycin. Resistance rates among these *E. coli* strains included 83% for gentamicin, 75% for ampicillin, 65.3% for trimethoprim, 55.8% for oxytetracycline, 36.5% for chloramphenicol, 30.7% for norfloxacin, and 26.9% for nalidixic acid. Notably, 62.8% of the tested isolates remained sensitive to ciprofloxacin.

In Egypt, according to Elafify et al. [103], 36 STEC isolates recovered from milk and dairy products carried *stx1* and/or *stx2* genes, while 14 and 3 of those possessed the *eaeA* gene and the *rfbE* gene, respectively, exhibiting multidrug resistance. Approximately 86.11% of these isolates harbored extended-spectrum beta-lactamase encoding genes, specifically blaCTX-M-15, blaSHV-12, and blaCTX-M-14. Moreover, 33.33% of the isolates carried the plasmid-mediated quinolone resistance gene qnrS.

Given that pathogenic *E. coli* is associated with increased illness and mortality rates, assessing antimicrobial resistance profiles should be considered a crucial component of *E. coli* surveillance in food safety and public health laboratories across Arab countries. Studies in Arab countries, such as Algeria [104,105] and Iraq [106,107], have also shown antibiotic resistance in pathogenic E. coli isolated from food.

It is imperative to recognize that the extent of resistance serves as an informative indicator of the selection pressure resulting from antibiotic use and resistance issues in pathogens. Indiscriminate antibiotic use must be curtailed, as antibiotics may lose their effectiveness against pathogens, particularly since *E. coli* acquires antibiotic resistance at a faster rate than most other bacteria [18]. The escalating global prevalence of antibiotic resistance is a matter of significant concern. It is widely acknowledged that the primary driver of resistance development in pathogenic bacteria is the excessive use of antibiotics [108]. This prevailing situation has facilitated the emergence and dissemination of antibiotic-resistant bacteria and resistance genes. Antibiotic resistance can stem from antibiotic use for treatment in both humans and animals, as well as from prophylactic and growth-promoting antibiotic use in animals [18].

## 4. Conclusions

Foodborne infections originating from bacterial pathogens like pathogenic *E. coli* are a prevalent cause of human illnesses in the Arab world, leading to significant economic losses and public health consequences. These *E. coli* pathogens’ genetic material is frequently found in various food items across Arab countries. The existing evidence highlighted in this review emphasizes that the identification of these bacterial pathogens is common in animal-based food products. In contrast, when it comes to fruits and vegetables, the available data on these pathogens are limited compared to animal-derived foods. These bacteria can enter the human food supply chain from their initial production stages to the final consumption of products. The emergence of drug-resistant strains has raised serious concerns about public health regarding these bacterial pathogens. Despite some reports on the prevalence of foodborne bacteria in animal-based foods, livestock, and humans, the extent of these pathogens in animal-based foods within the Arab region remains insufficiently studied. The associated risk factors are not well defined, and there is a lack of comprehensive documentation on human infections resulting from foodborne exposure. This literature review underscores the persistent challenge posed by *E. coli* pathogens to food safety and public health in the Arab world. Consequently, we propose the following recommendations: establish a coordinated surveillance and monitoring system for foodborne pathogens and their antimicrobial resistance at the national and regional levels across Arab countries to develop informed control and prevention strategies against these pathogens; generate epidemiological data on risk factors and the incidence of human infections linked to foodborne illnesses, focusing on national-level documentation; raise public awareness based on scientific risk analysis of bacterial pathogens responsible for foodborne infections; and employ advanced molecular-level characterization techniques, such as whole-genome sequencing, to guide the implementation of improved prevention and control strategies throughout Arab countries.

## Data Availability

The data used to support the findings of this study can be made available by the corresponding author upon request.

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
