# Peer review of "Pathogenic E. coli in the Food Chain across the Arab Countries: A Descriptive Review"

_foods, 2023, doi:10.3390/foods12203726_

Round 1

Reviewer 1 Report

This article aimed to offer insights into recent literature regarding the occurrence of harmful E. coli bacteria in the food supply of Arab countries. Additionally, it aimed to summarize existing information on health issues and the state of antibiotic resistance.

The article is straightforward, and it contains original information. However, this article would be improved if the authors revised and clarified the following: 

Line 26. Data presented in the year 2012 is not recent times. If it is available, please provide the data with more recent ones.

Lines 31-32 and 37-38. Why did the authors choose E. coli as a prevalence assessment study out of the disease-causing pathogens mentioned here?

Lines 79, 80, 91, and 95. The information presented in these sentences is redundant and can be combined in one sentence.

Lines 79-95. Are the contents described necessary for the scope of this review?

Line 135. Remove “infecting.”

Line 147. Remove “Enteropathogenic E. coli or.”

Line 152. Revise to “Ferhat et al. [31,32] successfully ….”

Lines 165, 177-178, 241-242. Remove “Enteropathogenic E. coli.”

Lines 155-157, 165-167, 184-186, 196-198, 227-229, 236-238, 261-262, 272-273, 277-280. These sentences may be combined and presented as one representative monitoring and control measure.

Lines 181-183. Revise to “sausage” and “luncheon.”

Lines 189. Italicize to “eaeA” for consistency.

Lines 248-250. Remove the sentence.  

Line 95. Remove “(EIEC).”

Line 311. May add more references related to the studies.

Lines 322 and 324. Italicize to “ial.”

Lines 348-350. Revise to “(50/20, 41.6%), (52/120, 43.3%), (47/120, 39.1%), (45/120, 37.5%), and (46/120, 38.3%),” accordingly.

Lines 353-353 and 365. Remove “Enterotoxigenic E. coli.”

Lines 364-365. Revise to “including Algeria [34] and Egypt [41].”

Lines 378-379. Remove “Enterohemorrhagic E. coli 378 O157:H7.”

Lines 390-391. Revise to “… of EHEC capable of …”

Line 394. Remove “in beef products.”

Lines 406, 413 and 414. Remove “hemolytic uremic syndrome.”

Lines 421-422 and 481-482. Remove “Shiga toxin-producing E. coli.”

Line 450. Revise to “antibiotic.”

Lines 451-453. You may need a citation for this sentence.

Line 453. Revise to “avian pathogenic E. coli (APEC).”

Lines 463-464. Please define “moderate resistance and low resistance.”

Line 481-484.  Revise to “In Egypt, according to Elafify et al. [103], thirty-six STEC isolates recovered from milk and dairy products carried stx1 and/or stx2 genes, while 14 and 3 of those possessed the eaeA gene and the rfbE gene, respectively,  exhibiting multidrug resistance.”

Lines 491-491. “Revise to “in Arab countries, such as Algeria [104,105] and Iraq [106,107], have also shown antibiotic resistance in pathogenic E. coli isolated from food.”

Lines 496-497. Add a citation.

Line 505. Revise to “bacterial pathogens like pathogenic E. coli are …”

Lines 521-522. Revise to “foodborne pathogens and their antimicrobial resistance at the national …”

Author Response

Comments and Suggestions for Authors:

This article aimed to offer insights into recent literature regarding the occurrence of harmful E. coli bacteria in the food supply of Arab countries. Additionally, it aimed to summarize existing information on health issues and the state of antibiotic resistance.

The article is straightforward, and it contains original information. However, this article would be improved if the authors revised and clarified the following: 

Line 26. Data presented in the year 2012 is not recent times. If it is available, please provide the data with more recent ones.

Corrected and reference changed.

Lines 31-32 and 37-38. Why did the authors choose E. coli as a prevalence assessment study out of the disease-causing pathogens mentioned here?

  • Common Occurrence: Escherichia coli ( coli) is a commonly occurring bacterium, and some strains are naturally found in the human intestines. Others, however, can be pathogenic and cause foodborne illnesses. This makes it a relevant target for prevalence assessment studies, especially in food safety and public health research.
  • Health Impact: Pathogenic strains of coli, such as E. coli O157:H7, can have severe health consequences, including food poisoning and severe gastrointestinal illnesses. Assessing the prevalence of these pathogenic strains is crucial for understanding public health risks.
  • Food Contamination: coli can be associated with food contamination, especially in cases of improper food handling or hygiene practices during food production, processing, or preparation. Studying its prevalence helps identify potential sources of contamination.
  • Indicator Organism: In some cases, coli is used as an indicator organism. High levels of E. coli may indicate poor hygiene, inadequate sanitation, or potential fecal contamination in food or water, warranting further investigation.
  • Regulatory Considerations: Many regulatory agencies worldwide have established guidelines and standards for coli levels in food products and water. Assessing its prevalence is essential for ensuring compliance with these regulations.

Lines 79, 80, 91, and 95. The information presented in these sentences is redundant and can be combined in one sentence.

Combined in one sentence.

Lines 79-95. Are the contents described necessary for the scope of this review?

This information serves to provide a comprehensive overview of E. coli, including its characteristics, taxonomy, and historical background.

Line 135. Remove “infecting.”

Removed

Line 147. Remove “Enteropathogenic E. coli or.”

Removed

Line 152. Revise to “Ferhat et al. [31,32] successfully ….”

Revised

Lines 165, 177-178, 241-242. Remove “Enteropathogenic E. coli.”

Removed

Lines 155-157, 165-167, 184-186, 196-198, 227-229, 236-238, 261-262, 272-273, 277-280. These sentences may be combined and presented as one representative monitoring and control measure.

Thank you for this correction.

Combined and presented as one representative monitoring and control measure.

Lines 181-183. Revise to “sausage” and “luncheon.”

Revised

Lines 189. Italicize to “eaeA” for consistency.

Done.

Lines 248-250. Remove the sentence. 

Removed.

Line 295. Remove “(EIEC).”

Removed.

Line 311. May add more references related to the studies.

Done.

Lines 322 and 324. Italicize to “ial.”

Done.

Lines 348-350. Revise to “(50/20, 41.6%), (52/120, 43.3%), (47/120, 39.1%), (45/120, 37.5%), and (46/120, 38.3%),” accordingly.

Done.

Lines 353-353 and 365. Remove “Enterotoxigenic E. coli.”

Done.

Lines 364-365. Revise to “including Algeria [34] and Egypt [41].”

Thank you

Done.

Lines 378-379. Remove “Enterohemorrhagic E. coli 378 O157:H7.”

Done.

Lines 390-391. Revise to “… of EHEC capable of …”

Done.

Line 394. Remove “in beef products.”

Done.

Lines 406, 413 and 414. Remove “hemolytic uremic syndrome.”

Done.

Lines 421-422 and 481-482. Remove “Shiga toxin-producing E. coli.”

Done.

Line 450. Revise to “antibiotic.”

Done.

Lines 451-453. You may need a citation for this sentence.

Added.

Line 453. Revise to “avian pathogenic E. coli (APEC).”

Done.

Lines 463-464. Please define “moderate resistance and low resistance.”

Definition: Moderate resistance refers to a level of resistance exhibited by a microorganism where it shows some reduced susceptibility to the antimicrobial agent, but the agent may still have some effectiveness in inhibiting or killing the microorganism.

Definition: Low resistance indicates that the microorganism exhibits a minimal level of resistance, and the antimicrobial agent remains highly effective in inhibiting or killing the microorganism.

Line 481-484.  Revise to “In Egypt, according to Elafify et al. [103], thirty-six STEC isolates recovered from milk and dairy products carried stx1 and/or stx2 genes, while 14 and 3 of those possessed the eaeA gene and the rfbE gene, respectively, exhibiting multidrug resistance.”

Done.

Lines 491-491. “Revise to “in Arab countries, such as Algeria [104,105] and Iraq [106,107], have also shown antibiotic resistance in pathogenic E. coli isolated from food.”

Done.

Lines 496-497. Add a citation.

Done.

Line 505. Revise to “bacterial pathogens like pathogenic E. coli are …”

Done.

Lines 521-522. Revise to “foodborne pathogens and their antimicrobial resistance at the national …”

Done.

Reviewer 2 Report

A very interesting manuscript on the prevalence of pathogenic E. coli in the food chain of the Arab countries. The manuscript is loaded with useful information; attention is needed in the following:

1.       the whole text is based on the occurrence of virulence genes in the genome of E. coli. This is not clear throughout the text. For example, in lines 152-167, 172-174 etc. the existence of studies on the detection of genes in some food samples is indicated. Although such studies do exist, they always highlight that the rest of the carrying bacterium is also present. A gene by itself cannot be pathogenic, it still needs the rest of the cell. This is not clear in the present manuscript and such a misunderstanding is very likely to happen. Please make sure that this is clarified throughout the text. In addition, failure to detect a gene may indicate its absence or the inability of the method employed to detect it. Since PCR is the method of choice in most of the cases, inability to detect a gene should lead to modifications of the protocol (including different sets of primers) that should be reported by the authors. Such modifications were not reported in the cited studies; therefore, the credibility of the results is, at least, questionable. Since the whole manuscript relied on presence or absence of genes, a comment stating this source of doubt should be made.

2.       l. 79-112. There are repetitions (eg. l. 80 & 90, l. 90 & 95, l. 106 & 112) and the paragraphs are not unified into one text. Please remove the repetitions and unify the text.

3.       l. 91. word ‘gram’ should be written with capitalized first letter

4.       All species and gene names should be written in italics. Especially in the case of genes, the first letter should not be capitalized (e.g. l. 89, 150, 333, references)

5.       l. 131. paragraph ‘2’ seems to be missing

6.       l. 150. what ‘94’ stands for?

7.       l. 180-183. An assumption of a possible correlation between food source and E. coli genetic content is made. This assumption refers to genes that do not give any obvious advantage to the extra-host lifestyle but contribute to the intra-host lifestyle of the bacterium. Please provide more evidence or rephrase.

8.       l. 191. is there evidence that local isolates do not harbor the gene?

9.       please apply the aforementioned corrections to the whole text and not only to the examples mentioned

Author Response

Comments and Suggestions for Authors:

A very interesting manuscript on the prevalence of pathogenic E. coli in the food chain of the Arab countries. The manuscript is loaded with useful information; attention is needed in the following:

1- The whole text is based on the occurrence of virulence genes in the genome of  coli. This is not clear throughout the text. For example, in lines 152-167, 172-174 etc. the existence of studies on the detection of genes in some food samples is indicated. Although such studies do exist, they always highlight that the rest of the carrying bacterium is also present. A gene by itself cannot be pathogenic, it still needs the rest of the cell. This is not clear in the present manuscript and such a misunderstanding is very likely to happen. Please make sure that this is clarified throughout the text.

Thank you very much for this good correction.

Done

In addition, failure to detect a gene may indicate its absence or the inability of the method employed to detect it. Since PCR is the method of choice in most of the cases, inability to detect a gene should lead to modifications of the protocol (including different sets of primers) that should be reported by the authors. Such modifications were not reported in the cited studies; therefore, the credibility of the results is, at least, questionable. Since the whole manuscript relied on presence or absence of genes, a comment stating this source of doubt should be made.

Done.

2- 79-112. There are repetitions (eg. l. 80 & 90, l. 90 & 95, l. 106 & 112) and the paragraphs are not unified into one text. Please remove the repetitions and unify the text.

Done.

3- 91. word ‘gram’ should be written with capitalized first letter

Done.

4- All species and gene names should be written in italics. Especially in the case of genes, the first letter should not be capitalized (e.g. l. 89, 150, 333, references)

Done.

5- 131. paragraph ‘2’ seems to be missing

Paragraph removed.

6- 150. what ‘94’ stands for?

Removed.

7- 180-183. An assumption of a possible correlation between food source and E. coli genetic content is made. This assumption refers to genes that do not give any obvious advantage to the extra-host lifestyle but contribute to the intra-host lifestyle of the bacterium. Please provide more evidence or rephrase.

Done.                                                                       

8- 191. is there evidence that local isolates do not harbor the gene?

There is limited information available regarding the presence or absence of the eaeA gene in E. coli isolates from Iraq.

9- please apply the aforementioned corrections to the whole text and not only to the examples mentioned

Done.

Round 2

Reviewer 2 Report

the author have addressed the comments

Author Response

I would like to thank the reviewer for his/her comments and suggestions, and it is good to hear that we fulfilled the enquiries raised earlier.
